# De Novo Synthesis of Poly(3-hydroxybutyrate-*co*-3-hydroxypropionate) from Oil by Engineered *Cupriavidus necator*

**DOI:** 10.3390/bioengineering10040446

**Published:** 2023-04-06

**Authors:** Mengdi Li, Wei Li, Tongtong Zhang, Keyi Guo, Dexin Feng, Fengbing Liang, Chao Xu, Mo Xian, Huibin Zou

**Affiliations:** 1State Key Laboratory Base of Eco-Chemical Engineering, College of Chemical Engineering, Qingdao University of Science and Technology, Qingdao 266042, China; annelimengdi@163.com (M.L.); 2021224065095@stu.scu.edu.cn (W.L.); zttyqz@outlook.com (T.Z.); gkyqust@163.com (K.G.); 2CAS Key Laboratory of Bio-Based Materials, Qingdao Institute of Bioenergy and Bioprocess Technology, Chinese Academy of Sciences, Qingdao 266101, China; fengdx@qibebt.ac.cn (D.F.); liangfb@qibebt.ac.cn (F.L.); xuchao@qibebt.ac.cn (C.X.); xianmo@qibebt.ac.cn (M.X.)

**Keywords:** biopolyester, poly(3-hydroxypropionate-*co*-3-hydroxybutyrate), engineered *Cupriavidus necator*, de novo biosynthesis, oil substrates

## Abstract

Poly(3-hydroxybutyrate-*co*-3-hydroxypropionate) [P(3HB-*co*-3HP)] is a biodegradable and biocompatible polyester with improved and expanded material properties compared with poly(3-hydroxybutyrate) (PHB). This study engineered a robust malonyl-CoA pathway in *Cupriavidus necator* for the efficient supply of a 3HP monomer, and could achieve the production of [P(3HB-*co*-3HP)] from variable oil substrates. Flask level experiments followed by product purification and characterization found the optimal fermentation condition (soybean oil as carbon source, 0.5 g/L arabinose as induction level) in general consideration of the PHA content, PHA titer and 3HP molar fraction. A 5 L fed-batch fermentation (72 h) further increased the dry cell weight (DCW) to 6.08 g/L, the titer of [P(3HB-*co*-3HP)] to 3.11 g/L and the 3HP molar fraction to 32.25%. Further improving the 3HP molar fraction by increasing arabinose induction failed as the engineered malonyl-CoA pathway was not properly expressed under the high-level induction condition. With several promising advantages (broader range of economic oil substrates, no need for expensive supplementations such as alanine and VB_12_), this study indicated a candidate route for the industrial level production of [P(3HB-*co*-3HP)]. For future prospects, further studies are needed to further improve the strain and the fermentation process and expand the range of relative products.

## 1. Introduction

With the exhaustion of fossil resources and environmental issues with regards to petro-derived durable polyesters, polyhydroxyalkanoates (PHA_S_) and other biodegradable polyesters have attracted interest as renewable substitutes [1]. One of the representative PHAs is poly(3-hydroxybutyrate) (P3HB) [2], which can be naturally produced by many microorganisms. *Cupriavidus necator* H16 (also called *Ralstonia eutropha* H16) is one of the commercial strains in P3HB production, which can efficiently produce intracellular P3HB [3] from oil substrates. The full genome study of *Cupriavidus necator* H16 revealed its efficient P3HB pathway, mainly including 3-ketothiolase (PhaA), NADPH-dependent acetoacetyl-CoA reductase (PhaB) and PHA synthase (PhaC) [3].

Due to the high crystallinity and brittleness, P3HB has limitations in the polyester market and hundreds of copolymer derivatives of P3HB have been developed to improve the material’s properties [4,5,6]. For example, the addition of 3-hydroxypropionate (3HP) monomer in poly(3-hydroxybutyrate-*co*-3-hydroxypropionate) (P(3HB-*co*-3HP)) will improve the thermal property, stiffness, malleability and tensile strength compared with the properties of P3HB [7]. Differently to 3HB, biosynthetic pathways (pduP pathway, malonyl-CoA pathway, β-alanine pathway) of 3HP do not exist in natural strains, thus feeding 3HP or the precursors of 3HP (1,5-pentanediol and β-alanine) during the fermentation process in the general strategy for the microbial production of P(3HB-*co*-3HP) [8,9,10,11].

A pioneering study attempted the de novo biosynthesis of P(3HB-co-3HP) from different carbon sources. For example, the pduP pathway was engineered in *Escherichia coli* for P(3HB-*co*-3HP) production from glycerol [12]. Another study further engineered the pduP pathway in *E. coli* to produce P(3HB-*co*-3HP) from glucose [13]. However, the major bottleneck of the pduP pathway is that vitamin B12 is required to maintain the function of glycerol dehydratase. The malonyl-CoA pathway was also engineered in *C. necator* for the de novo production of P(3HB-*co*-3HP) from fructose or alkanoic acids; however, due to the low efficiency of malonyl-CoA reductase under a mild temperature, the content of 3HP monomer was quite low (around 2% mol of 3HP) in P(3HB-*co*-3HP) [14], indicating the low efficiency of the natural malonyl-CoA pathway in the engineered strains.

Therefore, it is essential to further research and optimize the de novo biosynthesis of P(3HB-*co*-3HP) from a broader range of carbon sources by using the malonyl-CoA pathway. In this study, a modified full malonyl-CoA pathway was engineered in *C. necator* for the de novo biosynthesis of P(3HB-*co*-3HP) from three oil substrates. In addition, this study researched the effects of induction level (arabinose) on the production of P(3HB-*co*-3HP), aiming to regulate the content of 3HP in polyester products.

## 2. Materials and Methods

### 2.1. Strains and Plasmids

Detailed information of strains and plasmids used in this study is summarized in Table 1. Three genes of key enzymes (*pct*, propionate CoA-transferase; *mcr* (1–549), N-terminal of malonyl-CoA reductase; *mcr* (550–1219), C-terminal of malonyl-CoA reductase; *acc ADBC*, ATP-dependent acyl-CoA synthetase) were cloned from donor plasmids into backbone vector pBBR1MCS-2 by several steps (Figure 1); in addition, the arabinose operon (*ara C-P_BAD_*) was cloned from *E. coli* BL21 genome into backbone vector to gain the final expression vector pBBR-ara-pct-mcr-acc ADBC. The detailed cloning process is summarized in Figure 1 and the primer information is summarized in Table 2.

The pBBR-ara-pct-mcr-acc ADBC was firstly transformed into *E. coli* S17 before its conjugational transfer into the chassis strain *C. necator* H16 to gain *C. necator* ZL1 for PHA production. During cloning, transformation and conjugation process, *E. coli* strains were cultivated at 37 °C in Luria–Bertani (LB) broth (10  g/L peptone, 5  g/L yeast extract and 10  g/L NaCl) unless specified; *C. necator* strains were cultivated in a nutrient-rich Tryptic Soy Broth (TSB), which contains 17 g/L peptone, 3 g/L soy peptone, 5 g/L NaCl, 2.5 g/L K_2_HPO_4_. Kanamycin (50 µg/mL for *E. coli* strains and 150 µg/mL for *C. necator* strains) or gentamicin (10 μg/mL for *C. necator* strains) were added in the medium when necessary.

### 2.2. P(3HB-co-3HP) Fermentation and Purification

To obtain first seed culture, a single colony of *C. necator* ZL1 was inoculated into 10 mL of TSB medium in a 50 mL centrifuge tube supplemented with 150 µg/mL kanamycin and 10 μg/mL gentamicin at 220 rpm for 24 h at 30 °C. Under inoculation-to-flask volume ratio of 1/20, the second seed culture was inoculated into 100 mL of TSB medium in a 200 mL shake flask with kanamycin and gentamicin at 220 rpm for 16 h at 30 °C.

For flask level fermentation, 10 mL second seed culture was inoculated (inoculation ratio 1/20) in total 200 mL MS medium with kanamycin and gentamicin and cultivated at 30 °C, 220 rpm in Erlenmeyer flasks (baffled base). MS medium contained 9.0 g/L Na_2_HPO_4_·12H_2_O, 1.5 g/L KH_2_PO_4_, 1 g/L (NH_4_)_2_SO_4_, 0.4 g/L MgSO_4_·7H_2_O, 0.031 g/L FeCl_3_, 0.04 g/L citric acid, 0.015 g/L CaCl_2_·2H_2_O and 0.1 mL/L storage solution of trace elements (each 100 mL storage solution of trace elements containing 0.10 g ZnSO_4_·7H_2_O, 0.30 g H_3_BO_3_, 0.01 g CuSO_4_·5H_2_O, 0.03 g MnCl_2_·4H_2_O, 0.20 g CoCl_2_·6H_2_O, 0.02 g NiCl_2_·6H_2_O); total 5 g/L marketing soybean oil (Wilmar Inc., Great World City, Singapore), palm oil (Wenling Inc., Qingdao, China) or olive oil (Wilmar Inc., Great World City, Singapore) was individually utilized as carbon source (total 1 g oil substrate in 200 mL medium). P(3HB-*co*-3HP) production was induced by the addition of different levels of (0.25, 0.5, 1.0 or 2.0 g/L) arabinose at 12 h after inoculation. At 72 h after inoculation, the fermentation was ended for product extraction and characterization.

For fed-batch level fermentation, 100 mL second seed culture was inoculated in 2 L MS medium with kanamycin and gentamicin in a 5 L bioreactor (Bailun Inc., Shanghai China). Total 15 g/L (fed rate round 0.3 g/L/h) marketing soybean oil (Wilmar Inc., Great World City, Singapore) was fed during fermentation process (72 h). The fermentation process was operated under the following conditions: temperature 30 °C, pH was controlled at 7.0 ± 0.1 by automatic addition of 5 mol/L KOH, the aeration rate was controlled at one vvm, and dissolved oxygen (DO) level was maintained at 20 ± 1% by adjusting the stirring rate. The PHA production was induced by the addition of 0.5 g/L arabinose at 12 h after inoculation. At 72 h after inoculation, the fermentation was ended for product extraction and characterization.

After fermentation, the broth was centrifuged at 12,000 rpm for 20 min at 4 °C. The collecting cells were washed by ethanol once and then by distilled water twice before lyophilization. Polyester products were extracted from the lyophilized cells with hot chloroform in a Soxhlet apparatus and filtered with a PTFE membrane filter to remove the cell debris and finally purified by reprecipitation with ice-cold ethanol [17].

### 2.3. Analytical and Statistical Methods

Cell densities of the cultures were determined by measuring optical density at 600 nm using a spectrophotometer. Cell density samples were diluted as necessary so as to fall within the linear range. DCW (dry cell weight, g/L) of each sample was recorded after lyophilization. PHA wt% (weight percentage of PHA product towards DCW) was calculated after PHA extraction.

The polyester product was qualitatively detected by ^1^H NMR analysis (Bruker AM-500 MHz NMR spectrometer) in CDCl_3_ solution. Molecular weight (Mw) of polyester product was determined via gel permeation chromatography (GPC), with dichloromethane as moving phase at a flow rate of 0.35 mL/min, and 1.0 mg/mL of polyester solution was applied for GPC analysis [18]. PHA molar fraction and composition of each sample was roughly estimated via the comparison of the proton peak areas of featured polymer units by ^1^H NMR [9], and then accurately determined by gas chromatography [19,20]. During GC analysis, sulfuric acid–methanol solution was applied for the methanolysis of sample. A total of 15 mg of sample or standards (3HP or 3HB) was dissolved in methanolysis solution (1.7 mL methanol, 0.3 mL sulfuric acid and 2 mL chloroform) and heated at 100 °C for 150 min. After methanolysis, 1 mL distilled water was mixed to cool the sample to room temperature and 0.5 μL of organic layer was transferred for GC assay (SHMADZU GC-2010 Pro gas chromatograph equipped with an InertCap 1 capillary column and FID). The program was set at 80 °C for 1 min, 15 °C/min to 160 °C, at 160 °C for 3 min, 10 °C/min to 180 °C, and at 180 °C for 5 min. The temperatures of injector and detector were held at 230 °C and 250 °C, respectively. N_2_ was used as the carrier gas at the flow rate of 30 mL/min.

The shown data are means of three repeated fermentations together with their standard deviation. The significance of differences between mean values of samples was compared by Student’s *t*-test. Differences at *p* < 0.05 were considered obvious and those at *p* < 0.01 were considered significant.

## 3. Results and Discussion

### 3.1. Engineering the Fragmentized MCR in C. necator Promoted P(3HB-co-3HP) Production in Mild Condition

As shown in Figure 2, a full malonyl-CoA pathway was successfully engineered in *C. necator* ZL1 for P(3HB-*co*-3HP) production. Compared with the natural malonyl-CoA pathway [21,22], *C. necator* ZL1 utilized an engineered malonyl-CoA reductase (MCR), which was derived from the N- and C-terminal of natural MCR [16]. In 3HP production, the fragmentized MCR presented a higher efficiency in the mild condition (around 30 °C) than the natural MCR, which preferred a higher temperature (57–59 °C) [16,23]. In this study, the advantage of fragmentized MCR in P(3HB-*co*-3HP) production was also approved. The increased incorporation of 3HP into P(3HB-*co*-3HP) was confirmed by qualitative NMR analysis (Figure 3) and a quantitative GC assay in flask fermentation by *C. necator* ZL1 under 30 °C; the mol% of 3HP in the copolymer can reach 20–30% (Table 3), which is significantly higher than in the previous study that utilized natural MCR and only produced a copolymer with 2% (mol%) of 3HP [14]. Other than MCR, propionate CoA-transferase (PCT) is another important enzyme to generate 3HP-CoA before the polymerization of P(3HB-*co*-3HP) (Figure 2). An evolved PCT (V193A, from *Clostridium propionicum*) has previously confirmed its activities in the biosynthesis of PLA (poly-lactic acid) [15], and its CoA-transferase activity towards the 3HP substrate was confirmed in this study. PHA synthase is the last key enzyme in the biosynthesis of P(3HB-*co*-3HP) (Figure 2). This study did not overexpress any heterologous PHA synthase and confirmed that natural PHA synthase in the chassis strain can effectively polymerize both 3HP-CoA and 3HB-CoA.

However, when a larger amount of inducer (1.0 and 2.0 g/L of arabinose) was added to further promote the production of 3HP, the whole malonyl-CoA pathway was depressed in *C. necator* ZL1. The incorporation of 3HP was almost absent in the high-level induction groups (Table 3). Although arabinose can be utilized as a non-cytotoxic inducer for *C. necator* strains [24], it was demonstrated in this study that over-induction by a large dosage of arabinose had side-effects on the expression of the malonyl-CoA pathway. The detailed mechanism towards the inhibited malonyl-CoA pathway was unclear. It is speculated that one or several overexpressed enzymes in the malonyl-CoA pathway may have lost their activities for 3HP biosynthesis.

As indicated by a recent study [11], other candidate strategies should be tested to further improve the 3HP molar fraction in P(3HB-*co*-3HP): (1) the supplementation of cysteine (increase CoA supply); (2) the suppression of the 3HB pathway (by interrupting genes towards 3HB-CoA biosynthesis); (3) and the utilization of 3HP favored PHA synthases.

### 3.2. Production of P(3HB-co-3HP) from Variable Oil Substrates

Another advantage of the malonyl-CoA pathway in PHA production is that it can theoretically utilize any carbon source. Earlier studies have produced P(3HB-*co*-3HP) from variable carbon sources such as gluconate/alanine [11], glycerol [12], fructose and soybean oil (only trace 3HP incorporation in the copolymer) [14]. This study further confirmed that multiple oil substrates can be utilized as sole carbon sources in P(3HB-*co*-3HP) production (Table 3). From oil substrates, hydrolyzed fatty acids and glycerol can be converted to acetyl-CoA for the 3HB and 3HP pathway (Figure 2). *C. necator* strains have a strong ability to hydrolyze oil substrates, as indicated in an earlier study [25].

In comparing the effects of oil substrates on the production of P(3HB-*co*-3HP), the group of olive oil had the best cell growth (Table 3), which was slightly higher than that of the soybean oil group (*p* > 0.05) and obviously higher than that of the palm oil group (*p* < 0.05). However, in consideration of the titer of P(3HB-*co*-3HP), the soybean oil group had the highest titer, which was obviously higher than that of the olive oil group (*p* < 0.05) and significantly higher than that of the palm oil group (*p* < 0.01). The high PHA content (around 60%, Table 3) majorly contributed to the high titer of the soybean oil group. These results indicate that a broader range of oil substrates can be utilized for P(3HB-co-3HP) fermentation, and are similar to the results of studies in being able to produce PHA from oil substrate by *C. necator* strains [25].

### 3.3. Fed-Batch Production of P(3HB-co-3HP)

Of all the flask level studies (Table 3), the soybean oil group with 0.5 g/L arabinose induction exhibited better performance in the general consideration of the PHA content, PHA titer and 3HP molar fraction. Thus, this condition was chosen to cultivate *C. necator* ZL1 in a bioreactor to further investigate the scale-up results.

The fed-batch fermentation significantly improved the cell growth and titer (CDW reached 6.08 g/L, titer reached 3.11 g/L at 72 h) compared with flask level fermentation (*p* < 0.01, Table 3). The 3HP molar fraction reached 32.25%, which was slightly higher than that of flask level fermentation (27.57%), but the difference was not obvious (*p* > 0.05). The molecular weight of fed-batch-level-produced P(3HB-*co*-3HP) was detected by GPC. The results showed that the Mw (weight-average molecular weight) reached 201,105 (Da), the Mn (number-average molecular weight) reached 105,845 (Da), and the M_w_/M_n_ (molecular weight dispersion index, denotes the molecular weight distribution width of the polymer) was calculated as 1.9.

The comparison of current P(3HB-co-3HP) studies is summarized in Table 4. Compared with the engineered *E. coli* strains, *C. necator* strains have the advantage of utilizing a broader range of substrates, as approved in this study in that multiple oil substrates can be utilized as sole carbon sources for P(3HB-*co*-3HP) fermentation. Among all the studies utilizing *C. necator* strains, this study showed the best titer and a reasonable 3HP molar fraction. Another advantage of the proposed method in this study comes from the efficient malonyl-CoA pathway: no specific supplementation (alanine by β-alanine pathway; VB_12_ by pduP) is needed, thus reducing the cost for industrial applications. The disadvantage of the proposed method is that the 3HP molar fraction cannot be regulated and increased, and this demands further studies to engineer a more robust and under-regulation malonyl-CoA pathway in the chassis strain.

However, the bioreactor-scale fermentation in this study did not present a satisfactory biomass (maximum 6.08 g/L). One of the main bottlenecks was that the oil substrate was evenly fed during the fermentation process and the substrate’s concentration was not properly maintained to further improve the biomass accumulation. Only three oil substrates and several induction levels were tested in this study, and further systematic process optimization and improvement are required to achieve satisfactory cell growth and P(3HB-*co*-3HP) production.

## 4. Concluding Remarks and Future Prospects

By engineering a more efficient malonyl-CoA pathway in *C. necator*, this study solved the bottleneck question from a pioneering study [14] in that natural MCR presented low efficiency at a mild temperature (around 30 °C). Due to the increased efficiency of the malonyl-CoA pathway to supply 3HP, the 3HP molar fraction in P(3HB-*co*-3HP) was significantly increased (to 20–30 mol%) compared with the pioneering study [14]. However, when a higher induction level of arabinose was added in this study, the 3HP molar fraction was significantly decreased, indicating that the engineered malonyl-CoA pathway was not properly expressed in the high-induction condition. Compared with the β-alanine and pduP pathway [10,11,12], which achieved the production of P(3HB-co-3HP) with a high molar fraction (to 90 mol%), the malonyl-CoA pathway should be further investigated to improve its performance in the production of HP containing polyester.

In addition, this study confirmed that oil substrates can be utilized as a sole carbon source in P(3HB-*co*-3HP) production through the malonyl-CoA pathway. The utilization of oil substrates is normally seen in industrial PHA production by *C. necator* strains [25], and the production of P(3HB-*co*-3HP) from oil substrates is cost-effective as no additional supplementation is demanded, as seen in the production of P(3HB-*co*-3HP) through the β-alanine and pduP pathway [10,11,12]. The low-cost method of this study provided candidate techniques for tentative industrial applications.

Although the production of P(3HB-*co*-3HP) by *C. necator* through the malonyl-CoA pathway has several advantages (broader carbon source, no expensive supplementation such as alanine and VB_12_), the strain and process demand further optimization in future studies: (1) testing a broader range of substrates in P(3HB-*co*-3HP), such as waste or recycled cooking oil; (2) developing the metabolic engineering to gain adjustable metabolic flux towards 3HB and 3HP (to regulate 3HP molar fraction); (3) the expanding of product streamlines by *C. necator* through the malonyl-CoA pathway, such as P3HP (natural 3HB flux should be depleted) and other copolymers containing the 3HP monomer; and (4) the development of a high-density fermentation process to improve the titer and reduce the unit cost.

## Figures and Tables

**Figure 1 bioengineering-10-00446-f001:**
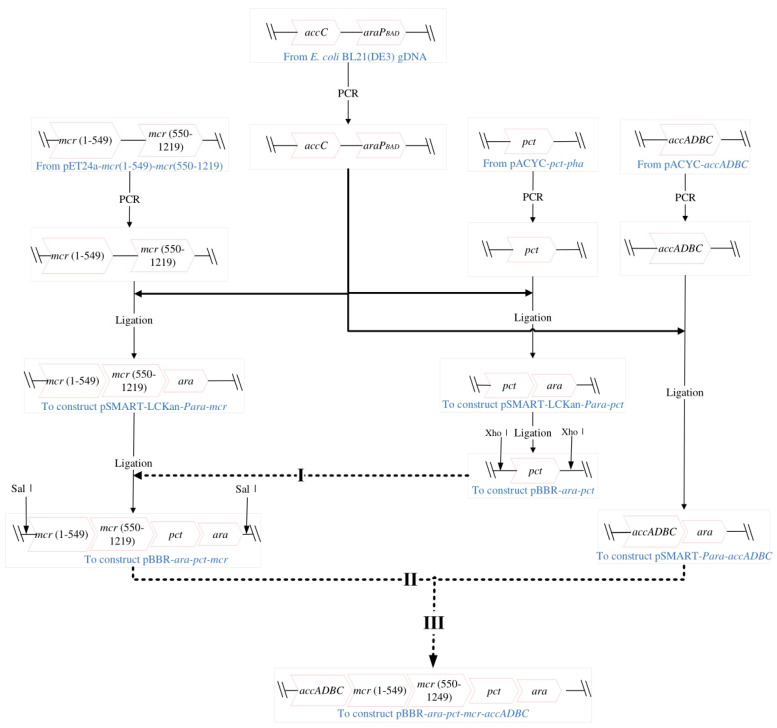
Cloning steps of targeting genes *ara C*−*P_BAD_*, *pct*, *mcr*(1−549), *mcr*(550−1219) and *acc ADBC* into final expression vector pBBR−*ara*−*pct*−*mcr*−*acc ADBC*. Firstly, targeting genes or fragments were cloned into cloning vectors pSMART−*Para*−*pct*, pSMART−*Para*−*mcr*, pSMART−*Para*−*acc ADBC*. Then intermediate vectors pBBR−*ara*−*pct*, pBBR−*ara*−*pct*−*mcr* and final expression vector pBBR−*ara*−*pct*−*mcr*−*acc ADBC* were constructed by homologous recombination. Cloning and testing primers were summarized in Table 2.

**Figure 2 bioengineering-10-00446-f002:**
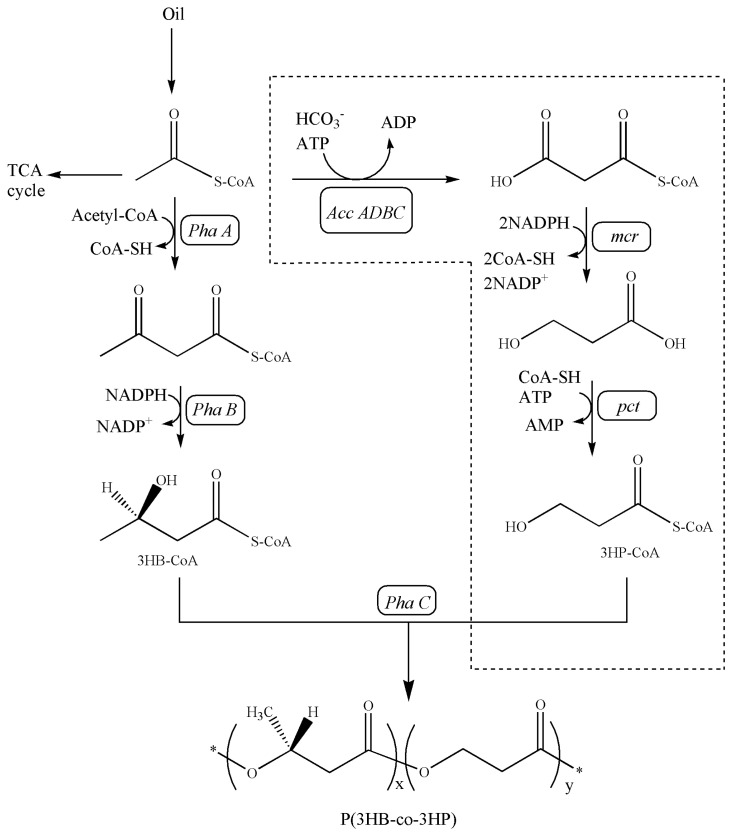
Artificial metabolic pathway for the de novo biosynthesis of P(3HB−*co*−3HP) copolymer from oil by *C. necator* ZL1. *Acc ADBC*, acetyl−CoA carboxylase; *mcr*, malonyl−CoA reductase; *pct*, propionyl−CoA transferase; PhaA, β-ketothiolase; PhaB, NADPH-dependent acetoacetyl−CoA reductase; PhaC, PHA synthase; *, repeating unit.

**Figure 3 bioengineering-10-00446-f003:**
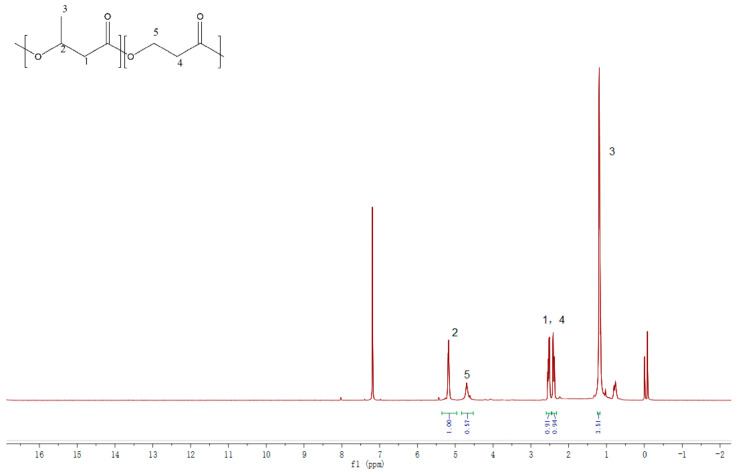
^1^H NMR spectrum of P(3HB−*co*−3HP) synthesized by *C. necator* ZL1. The representative resonance of the C_5_ methylene group in the 3HP monomer was detected at 4.70 ppm, confirming that the 3HP unit was polymerized to produce P(3HB-*co*-3HP). The resonance of the C_4_ methylene group in the 3HP unit (2.40 ppm) was detected near the resonance of the C_1_ methylene group in 3HB (2.50 ppm).

**Table 1 bioengineering-10-00446-t001:** Strains and plasmids used in this study.

Strains/Plasmids	Description	Source
**Strains**		
*Cupriavidus necator* H16 (*Ralstonia eutropha* H16)	Chassis strain	DSM 428
*Escherichia coli* S17	Donor strain for conjugational transfer	DSM 9079
*Escherichia coli* BL21(DE3)	F- ompT hsdSB(rB-mB-) gal dcm rne131(DE3)	Invitrogen
*Cupriavidus necator* ZL1	Engineered *Cupriavidus necator* H16 carrying pBBR-ara-pct-mcr-acc ADBC	This study
Plasmids		
pSMART-LCKan	Cloning plasmid	Lucigen
pBBR1MCS-2	Starting plasmid (broad host range)	Lab stored
pACYC-pct-pha	Carrying gene *pct*, evolved propionate CoA-transferase (V193A) from *Clostridium propionicum*	[15]
pACYC-acc ADBC	Carrying gene *acc ADBC*, acetyl-CoA carboxylase from *Escherichia coli* K-12	[16]
pET24a-mcr(1–549)-mcr(550–1219)	Carrying gene *mcr*(1–549) and *mcr*(550–1219), evolved N-terminal and C-terminal of malonyl-CoA reductase from *Chloroflexus aurantiacus*	[16]
pSMART-LCKan-Para-pct	pSMART-LCKan derivative carrying genes *ara*, *pct*	This study
pSMART-LCKan-Para-mcr	pSMART-LCKan derivative carrying genes *ara*, *mcr*	This study
pSMART-LCKan-Para-acc ADBC	SMART-LCKan derivative carrying genes *ara*, *acc* ADBC	This study
pBBR-ara-pct	pBBR1MCS-2 derivative carrying genes *ara*, *pct*	This study
pBBR-ara-pct-mcr	pBBR1MCS-2 derivative carrying genes *ara*, *pct*, *mcr*(1–549)-*mcr*(550–1219)	This study
pBBR-ara-pct-mcr-acc ADBC	pBBR1MCS-2 derivative carrying genes *ara*, *pct*, *mcr*(1–549)-*mcr*(550–1219), *acc ADBC*	This study

**Table 2 bioengineering-10-00446-t002:** Primers used in this study.

Primer Name	Sequence
ara F	ATATCAAGCTTGAATTCGTTTTATGACAACTTGACGGCTAC
ara R	CTTTGCGCATCGTTTCACTCCATCCAAAAAAAC
araP F	ATATCAAGCTTGAATTCGTTAAGAAACCAATTGTCCATATTGC
araP R	CTTTGCGCATCGTTTCACTCCATCCAAAAAAAC
accB F1	GAGACCTTAGGAGGTAAACATATGGATATTCGTAAGATTAAAAAACTGATC
accB R1	ATTCTCTGCAGGCCTGTACAGTTACTCGATGACGACCAG
accD F2	TGAGCTACGGTTACGCGTAAATGAGCTGGATTGAACGAATTAAAAGC
accD R2	GAATATCCATTCAGGCCTCAGGTTCCTGAT
accADBC F	CCTCCGACCGGAGGCTTTTAGCTGGAAGAAACCAATTGTCCATATTGCATCAG
accADBC R	TTCGATATCAAGCTTATCGATACCGAGTCAAAAGCCTCCGGTCGG
accBC F	TGAGGCCTGAATGGATATTCGTAAGATTAAAAAAC
accBC R	CGATATCTAGAGAATTCGTCTTATTTTTCCTGAAGACCGAG
mcr R1	GCTTATCGATACCGTCGACCAGTCAAAAGCCTCCGGTCGG
mcr F1	TGACTGTACCGGGCCCCCCCAAGAAACCAATTGTCCATATTGCATCAG
Mcr(1–549)F1	GGAGTGAAACGATGAGCGGAACAGGACGACTG
Mcr(550–1219)R1	CGATATCTAGAGAATTCGTCTTACACGGTAATCGCCCGTCCGCG
M13 F	GTAAAACGACGGCCAGT
M13 R	CAGGAAACAGCTATGAC
pct F	GAGTGAAACGATGCGCAAAGTGCCGATTATTAC
pct R	CGATATCTAGAGAATTCGTCTTATGATTTCATTTCTTTCAGGCCC
SL1	CAGTCCAGTTACGCTGGAGTC
SR1	GGTCAGGTATGATTTAAATGGTCAGT

**Table 3 bioengineering-10-00446-t003:** P(3HB-*co*-3HP) production by *C. necator* ZL1 under different conditions.

Level	Oil	Arabinose (g/L)	DCW (g/L)	PHA (wt%)	PHA (g/L)	3HP (mol%)
Flask	Soybean oil	0.25	1.52 ± 0.11	57.33 ± 5.11	0.87 ± 0.11	21.36 ± 4.11
Flask	Soybean oil	0.50	2.11 ± 0.13	62.42 ± 5.51	1.31 ± 0.12	27.57 ± 5.81
Flask	Soybean oil	1.00	1.56 ± 0.09	56.76 ± 4.51	0.89 ± 0.09	1.12 ± 0.05
Flask	Soybean oil	2.00	1.21 ± 0.11	52.41 ± 5.21	0.63 ± 0.06	0
Flask	Palm oil	0.50	1.56 ± 0.18	42.41 ± 6.21	0.66 ± 0.09	22.13 ± 6.05
Flask	Olive oil	0.50	2.35 ± 0.21	45.18 ± 4.11	1.06 ± 0.08	25.16 ± 5.15
Fermenter	Soybean oil	0.50	6.08 ± 0.51	51.15 ± 3.42	3.11 ± 0.29	32.25 ± 4.02

**Table 4 bioengineering-10-00446-t004:** Comparison of P(3HB-*co*-3HP) production by different strains.

Stain	Pathway	Substrate	Supplementation	Titer (g/L)	3HP (mol%)	Reference
*C. necator*	malonyl-CoA	oil	No	3.1	20–30	This study
*C. necator*	malonyl-CoA	sugar and oil	No	2.1	trace	[13]
*C. necator*	β-alanine	gluconate, alanine	cysteine	1.6	0–91	[10]
*E. coli*	pduP	glycerol	VB_12_	9.8	10–95	[11]
*E. coli*	pduP	glucose	VB_12_ (in LB)	2.1	60–80	[12]

## Data Availability

Not applicable.

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
