# Peer review of "De Novo Synthesis of Poly(3-hydroxybutyrate-co-3-hydroxypropionate) from Oil by Engineered Cupriavidus necator"

_bioengineering, 2023, doi:10.3390/bioengineering10040446_

Round 1
Reviewer 1 Report
The authors proposed a new 3HP supply route and succeeded in synthesizing P(3HB-co-3HP) copolymer from soybean oil using an engineered Cupriavidus necator. Moreover, despite the single carbon source of soybean oil, the 3HP fraction reached 32 mol%, achieving a very high value. This is a significant improvement from the previous studies and this paper deserves publication in this journal.
Below are minor comments and questions for this study.
1. Is glycerol, a constituent of triacylglyceride, a precursor of 3HP?
2. Why malonyl-CoA pathway was depressed when arabinose concentration was high?
3. Is it possible to synthesize P(3HB-co-3HP) without the use of inducers such as arabinose?
4. Is the P(3HB-co-3HP) synthesized this study a random copolymer? Have the authors measured the melting point of these copolymers?
5. In the manuscript, the authors use DCW and CDW. If these means are the same, the authors should use one or the other.
Author Response
Correspondence to Reviewer 1
Comment 1. Is glycerol, a constituent of triacylglyceride, a precursor of 3HP?
Response to comment 1. Thank you for the question. Yes, glycerol can be converted to acetyl-CoA for further 3HP or 3HB biosynthesis. A new description of this information is added in revised manuscript (Section 3.2., line 203).
Comment 2. Why malonyl-CoA pathway was depressed when arabinose concentration was high?
Response to comment 2. Good question. We cannot clearly find out the mechanism why high level arabinose inhibited the malonyl CoA pathway, we speculated that over-expressed enzymes may lost their functions in 3HP supply. A new sentence is added in the revised manuscript (Section 3.1, line 191).
Comment 3. Is it possible to synthesize P(3HB-co-3HP) without the use of inducers such as arabinose?
Response to comment 3. Thank you for the question. Induction of the malonyl-CoA pathway by arabinose is compulsory for 3HP supply. Our pioneering study (without induction, data not shown in this study) proved this.
Comment 4. Is the P(3HB-co-3HP) synthesized this study a random copolymer? Have the authors measured the melting point of these copolymers?
Response to comment 4. Thank you for the question. We did not measure the thermo-properties of the products. But earlier study demonstrated that P(3HB-co-3HP) produced by C. necator through malonyl-CoA pathway is random copolymer, as one glass-transition temperature was reported (Fukui et al., 2009).
Comment 5. In the manuscript, the authors use DCW and CDW. If these means are the same, the authors should use one or the other.
Response to comment 5. Thank you for the comment. CDW was corrected to DCW in revised manuscript.
Reviewer 2 Report
The Brief Report presents the results of constructing a new pathway for the biosynthesis of Poly(3-hydroxybutyrate-co-3-hydroxypropionate) [P(3HB-co-3HP)] in the bacterial strain Cupriavidus necator. The strain was tested during cultivation in flasks and in a fermentation apparatus.
The work is interesting and aimed at solving topical issues.
But there are a few questions and comments.
1. The statement of the research problem is a bit confusing.
Line 58 - 59 "Therefore, it is essential to further engineered the natural malonyl-CoA pathway for the de novo biosynthesis of P(3HB-co-3HP) from economic carbon sources…….". As far as I could understand, the subject of the study was the Cupriavidus necator strain. And it was in the strain that the biosynthesis pathway was constructed, and the bacterial strain was subsequently tested for its ability to grow on oily substrates and synthesize [P(3HB-co-3HP)]. In addition, there is insufficient information about the engineering of a new biosynthetic pathway. Therefore, I think the wording of the problem is incorrect. She misleads the reader.
Line 61 – 62 The rationale for the choice of substrate is given, but this information has nothing to do with the formulation of the research problem.
Line 63 – 64 "In addition, this study optimized the fermentation process to achieve efficient production of P(3HB-co-3HP)." - I do not agree with this statement. The fermentation process is complex and multifactorial, but apart from testing on different oils and three concentrations of the inductor, the Report contains no information.
Considering that this is a Brief Report, the authors should more clearly state the purpose of the research and should not attempt to set several objectives in the Brief Report.
2. Information should be added to the materials and methods, how was the carbon substrate concentration controlled during the fed-batch fermentation?
3. At what concentration of oil was fermentation carried out in flasks?
4. What oil concentration was maintained in the fermentation apparatus?
5. Line 61 – 62 It is not clear what the author meant? The fact that oils can be used as a carbon substrate has long been known. And this was shown by many other studies, not “These results….”. How does the reference to the 1998 article fit in with "...current industrial process.."? Literature references are not correct. The first refers to the 1998 review, the second to the study of the bacterial genome. These publications do not reflect the problems of industrial production of PHA.
6. What is the reason for such a low productivity of the process during biosynthesis in a fermenter fed with a substrate? 6 g/l of biomass can be obtained in a flask without resorting to the use of a fermenter.
7. The conclusion should be corrected.
Line 242 – 243 The fact that oils can be used as a carbon substrate has long been known. And many other studies have shown this. I recommend further reading the literature on this subject.
In conclusion, it is worth paying more attention to the concrete results obtained in the study, and not to future prospects. The conclusion should correspond to the tasks set and in a short form present the specific results received or achieved.
Author Response
Comment 1. The statement of the research problem is a bit confusing.
Line 58 - 59 "Therefore, it is essential to further engineered the natural malonyl-CoA pathway for the de novo biosynthesis of P(3HB-co-3HP) from economic carbon sources…….". As far as I could understand, the subject of the study was the Cupriavidus necator strain. And it was in the strain that the biosynthesis pathway was constructed, and the bacterial strain was subsequently tested for its ability to grow on oily substrates and synthesize [P(3HB-co-3HP)]. In addition, there is insufficient information about the engineering of a new biosynthetic pathway. Therefore, I think the wording of the problem is incorrect. She misleads the reader.
Line 61 – 62 The rationale for the choice of substrate is given, but this information has nothing to do with the formulation of the research problem.
Line 63 – 64 "In addition, this study optimized the fermentation process to achieve efficient production of P(3HB-co-3HP)." - I do not agree with this statement. The fermentation process is complex and multifactorial, but apart from testing on different oils and three concentrations of the inductor, the Report contains no information.
Considering that this is a Brief Report, the authors should more clearly state the purpose of the research and should not attempt to set several objectives in the Brief Report.
Response to comment 1. Thank you for the comment. The revised the manuscript reorganized the last paragraph of induction section (line 58-62). A clearer and shortened statement of research problem and objective was written, which focused on one objective “studying the feasibility of three oil substrates in [P(3HB-co-3HP)] production through malonyl-CoA pathway”.
Comment 2. Information should be added to the materials and methods, how was the carbon substrate concentration controlled during the fed-batch fermentation?
Response to comment 2. Thank you for the comment. New information was added in revised manuscript, which described the adding rate of oil substrate. However, it was hard to detect the residue oil during fermentation process, so the residue oil in fermenter was not detected during fermentation process.
Comment 3. At what concentration of oil was fermentation carried out in flasks?
Response to comment 3. Thank you for the comment. New information was added in revised manuscript, which described the adding method and total amount of oil substrate. However, it was hard to detect the residue oil, so the residue oil in flask was not detected during fermentation process.
Comment 4. What oil concentration was maintained in the fermentation apparatus?
Response to comment 4. Thank you for the comment. It was hard to detect the residue oil during fermentation process, so the residue oil in fermenter was not detected during fermentation process.
Comment 5. Line 61 – 62 It is not clear what the author meant? The fact that oils can be used as a carbon substrate has long been known. And this was shown by many other studies, not “These results….”. How does the reference to the 1998 article fit in with "...current industrial process.."? Literature references are not correct. The first refers to the 1998 review, the second to the study of the bacterial genome. These publications do not reflect the problems of industrial production of PHA.
Response to comment 5. Thank you for the comment. At the end of section 3.2., we revised the improper description and citation, cited a new research article to support the discussion.
Comment 6. What is the reason for such a low productivity of the process during biosynthesis in a fermenter fed with a substrate? 6 g/l of biomass can be obtained in a flask without resorting to the use of a fermenter.
Response to comment 6. Thank you for the comment. At the end of section 3.3, a new sentence was added to describe the reason of unsatisfied cell growth in fermenter level fermentation.
Comment 7. The conclusion should be corrected.
Line 242 – 243 The fact that oils can be used as a carbon substrate has long been known. And many other studies have shown this. I recommend further reading the literature on this subject.
In conclusion, it is worth paying more attention to the concrete results obtained in the study, and not to future prospects. The conclusion should correspond to the tasks set and in a short form present the specific results received or achieved.
Response to comment 7. Thank you for the comment. The first paragraph of conclusion section was revised to summarize the main outcomes and concludes, which corresponded to the research objectives set in introduction section